# TSPO PET Imaging as a Potent Non-Invasive Biomarker for Diffuse Intrinsic Pontine Glioma in a Patient-Derived Orthotopic Rat Model

**DOI:** 10.3390/ijms232012476

**Published:** 2022-10-18

**Authors:** Céline Chevaleyre, Dimitri Kereselidze, Fabien Caillé, Nicolas Tournier, Nagore G. Olaciregui, Alexandra Winkeler, Xavier Declèves, Benoit Jego, Salvatore Cisternino, Sylvain Auvity, Charles Truillet

**Affiliations:** 1BioMaps, Service Hospitalier Frédéric Joliot, CEA, CNRS, Inserm, University of Paris-Saclay, 4 Place du Général Leclerc, 91401 Orsay, France; 2Institut de Recerca, Sant Joan de Déu Hospital, Carrer de Sant Quinti, 77, 08041 Barcelona, Spain; 3Inserm UMR-S1144, University of Paris Cité, 75006 Paris, France; 4Hôpital Universitaire Necker-Enfants Malades, Assistance Publique des Hôpitaux de Paris, AP-HP, 149 Rue de Sèvres, 75015 Paris, France

**Keywords:** DIPG biomarker, TSPO-PET imaging, PDOX

## Abstract

Diffuse intrinsic pontine gliomas (DIPG), the first cause of cerebral pediatric cancer death, will greatly benefit from specific and non-invasive biomarkers for patient follow-up and monitoring of drug efficacy. Since biopsies are challenging for brain tumors, molecular imaging may be a technique of choice to target and follow tumor evolution. So far, MR remains the imaging technique of reference for DIPG, although it often fails to define the extent of tumors, an essential parameter for therapeutic efficacy assessment. Thanks to its high sensitivity, positron emission tomography (PET) offers a unique way to target specific biomarkers in vivo. We demonstrated in a patient-derived orthotopic xenograft (PDOX) model in the rat that the translocator protein of 18 kDa (TSPO) may be a promising biomarker for monitoring DIPG tumors. We studied the distribution of 18F-DPA-714, a TSPO radioligand, in rats inoculated with HSJD-DIPG-007 cells. The primary DIPG human cell line HSJD-DIPG-007 highly represents this pediatric tumor, displaying the most prevalent DIPG mutations, H3F3A (K27M) and ACVR1 (R206H). Kinetic modeling and parametric imaging using the brain 18F-DPA-714 PET data enabled specific delineation of the DIPG tumor area, which is crucial for radiotherapy dose management.

## 1. Introduction

Diffuse intrinsic pontine gliomas (DIPG) are highly infiltrative pediatric gliomas. They are the leading cause of death in children with brain tumors, representing 10 to 20% of all pediatric brain tumors [1]. The median age at diagnosis is 6.8 years, with a median overall survival of 11 months [2]. This poor prognosis is directly linked to the difficulty of finding an efficient treatment. Surgical resection is not an option due to the highly infiltrative nature and the localization of DIPG in the brainstem, which regulates critical system functions such as respiration [3]. The conservation of an intact blood–brain barrier renders systemic chemotherapeutic treatment challenging [4,5]. The most effective treatment that has proved to prolong progression-free survival and median overall survival is fractionated radiation therapy. However, radiotherapy improves the average survival by only a couple of months, which is not enough to overcome the fatal term of the disease [6]. Several ongoing preclinical and clinical trials with innovative treatments, such as immunotherapy, stem cell transplantation, and targeting therapies, associated with adjuvant therapy techniques such as ultrasound therapy are showing some promising results [7,8,9]. However, those therapies and novel radiotherapy strategies will not be approved for patient care without proper clinical evaluation. Biomarkers for DIPG are therefore needed to monitor treatment efficacy over time.

Several genetic mutations are identified in DIPG, the most prevalent being the mutation of the histone H3 (H3K27M) [10]. In its 2021 classification of central nervous system tumors, the WHO added H3K27-altered diffuse midline gliomas as a distinct type within pediatric high-grade gliomas [11]. Although the prognostic value of the two types of H3K27M mutations (H3F3A and HIST1H3B) has been proven, its utility in therapeutic response assessment is limited by the feasibility of biopsies [12]. Biopsies are challenging due to the localization of the tumor and can present a risk of altering the patient’s well-being. MRI remains therefore the gold standard for the diagnosis and follow-up of DIPG progression. Diagnostic criteria for DIPG on MRI are the observation of an intrinsic lesion T1-hypointense and T2-hyperintense with irregular margins and sparse to no enhancement after gadolinium administration. The lesions should typically involve more than 50% of the axial diameter of the pons [13]. Abnormality on the diagnostic fluid-attenuated inversion recovery (FLAIR) MRI is usually used to define the target of fractioned radiotherapy. MRI presents some limitations in defining the grade, the aggressiveness, and the real extent of tumors. Those limitations especially impact clinical trials with observations of increased or decreased FLAIR signal inaccurately associated with tumor progression or regression after radiotherapy or chemotherapeutic treatments (pseudo-progression or pseudo-response) [14].

In contrast to conventional radiological techniques, molecular imaging (MI), especially positron emission tomography (PET), offers the possibility to quantify biological processes at the cellular and subcellular levels. MI may therefore define the actual extent of tumors and treatment response on highly infiltrative tumors such as DIPG. There is an unmet need for a specific PET ligand for DIPG tumors. Several groups have investigated the potential uses of PET imaging to improve diagnosis and assess therapeutic efficacy. ^18^F-2 fluoro-2-deoxy-d-glucose (^18^F-FDG), the most used PET ligand, does not present sufficient specificity with the pathology due to the high surrounding glucose metabolism in the healthy brain [15,16]. ^18^F-PARPi targeting poly [ADP-ribose] polymerase 1 (PARP-1) expression in DIPG, ^11^C-Choline assessing choline kinase regulation, and ^18^F-fluoro-3′-deoxy-3’-L-fluorothymidine (^18^F-FLT) PET, which allows in vivo assessment of tumor proliferation, have shown mitigated results and did not translate into clinical practice [17,18].

Several preclinical and clinical trials have observed a high expression of the 18 kDa translocator protein (TSPO) in adult gliomas compared to the low TSPO expression in healthy brain regions [19,20,21]. TSPO expression by neoplastic cells is suggested as a reliable prognostic biomarker as it increases with histological tumor grade [22]. Those results show the potential of TSPO PET imaging of gliomas to stratify patients and detect tumor progression [23]. We hypothesized that TSPO could be a relevant biomarker of DIPG. We investigated ^18^F-DPA-714, a TSPO PET radioligand used in many clinical trials, as a potential biomarker evaluating the extent of DIPG tumors in a patient-derived orthotopic xenograft (PDOX) rat model established with HSJD-DIPG-007 primary DIPG human tumor cells.

## 2. Results

### 2.1. Orthotopic DIPG Rat Model Characterization

The in vitro protein screening of HSJD-DIPG-007 (DIPG007) cells by western blot revealed a TSPO expression (Figure 1A). After the orthotopic implantation of DIPG007 cells (4 weeks post-injection), brain lysates from tumor-inoculated brains present a high ex vivo TSPO expression in the back brain (brain stem and cerebellum) compared with sham rats’ back brains lysates (Figure 1C). While DIPG007 TSPO expression seems weaker compared with U87 TSPO overexpression in vitro, it appears to be sufficient to differentiate tumor inoculated brain tissue from healthy brain tissue. DIPG proliferation 4 weeks post-injection outreaches the pons as seen on the H&E staining and the associated ^18^F-DPA-714 autoradiography (Figure 1D,E).

TSPO expression is not only specific to the glioma cells but can be highly expressed after glial cell activation, especially in activated microglial cells and reactive astrocytes [23]. TSPO has become, in this context, one of the most representative markers for neuroinflammation imaging [24]. Glioma-associated microglia/macrophages (GAMs) are the main glial cells that may impact TSPO expression distribution after activation. They can express TSPO and contribute to the ^18^F-DPA-714 PET signal. Therefore, we labeled CD11b to assess the microglial stake of TSPO expression in this DIPG rat model. CD11b^+^-GAMs presence seemed associated with tumoral cell infiltration (Appendix A), although CD11b^+^ microglial cells can be found around the injection side in sham-injected animals (Appendix A). A notably high CD11b^+^-GAMs recruitment was observed in the area with the highest tumoral cell density (Appendix A). However, the diffuse ^18^F-DPA-714 fixation on the autoradiography is more similar to the pattern of the human mitochondria staining, revealing the presence of DIPG007 cells (Figure 1E,F). The presence of the DIPG tumor drastically accounts for the uptake of ^18^F-DPA-714 compared with the limited impact of inflammation induced in the sham-injected rat brains (Figure 1E versus Appendix A). Higher resolution images showed stronger TSPO colocalization with human mitochondria compared with CD11b+-GAMs (Figure 1G,H). Altogether, TSPO seems a suitable biomarker for DIPG PET imaging.

### 2.2. ^18^F-FDG and ^18^F-DPA-714 PET Imaging of DIPG

^18^F-FDG PET imaging is the most important clinical PET radioligand to assess glioma evolution over time [25]. In order to compare conventional FDG determination of glucose metabolism, assessed using ^18^F-FDG, with TSPO PET imaging, both PET scans were performed on the same animals (Figure 2A). TSPO PET imaging with ^18^F-DPA-714 revealed significantly increased ^18^F-DPA-714 uptake in the pons of DIPG-bearing rats, as expected according to the H&E DIPG staining. In comparison, ^18^F-FDG uptake in the pons was not different in the presence of DIPG (1.71 ± 0.17 SUV for DIPG-bearing rats versus 1.56 ± 0.18 SUV for sham rats, *p*-value = 0.24) (Figure 2C). Uptakes of ^18^F-FDG and ^18^F-DPA-714 did not correlate. Astonishingly, tumor-bearing rats showed higher ^18^F-FDG uptake in the whole brain, suggesting increased brain glucose metabolism compared with sham rats (Appendix A).

Mean ^18^F-DPA-714 SUVs in the pons of tumor-bearing rats were higher than in the pons of sham animals (0.49 ± 0.09 versus 0.33 ± 0.04 SUV; *p*-value = 0.022) (Figure 2B). The pons was the only brain region in which a significantly increased uptake of ^18^F-DPA-714 was observed (Appendix A).

Time–activity curves (TAC) measured in the striatum of tumor-bearing and sham rats were similar (Figure 3A), thus consolidating the use of the striatum as a pseudo reference region with no or negligible specific uptake of ^18^F-DPA-714. Conversely, TACs measured in the region with DIPG proliferation, i.e., the pons region, are significantly different (Figure 3A). Accumulation of ^18^F-DPA-714 was estimated for both groups by calculating the area under the tumor time-activity curve (AUC). AUC was expressed as SUV·min. ^18^F-DPA-714 uptake is doubled in the presence of TSPO expressing DIPG cells, as highlighted by the significant rise of the pons to striatum AUC ratios of the tumor-bearing rats (AUCr = 2.03 ± 0.22 for tumor-bearing rats versus 1.55 ± 0.24 for sham rats; *p*-value = 0.02) (Figure 3B).

### 2.3. Parametric PET Images of the Brain Distribution Volume of ^18^F-DPA-714

The Logan graphical analysis applied voxel-wise results in a parametric map for volumes of distribution (*V_T_*). *V_T_* is a measure of the overall uptake in the tissue of interest relative to the blood compartment at equilibrium [26]. Brain parametric maps of *V_T_* were generated for each rat by using individual image-derived whole blood input function (Appendix A). The average *V*_T_ in the pons of the tumor-bearing rats compared with the sham rats was twice higher (*V*_T_ = 0.29 ± 0.11 mL/cm^3^ for the DIPG group versus *V*_T_ = 0.15 ± 0.14 mL/cm^3^ for the sham group; Appendix A). This can be directly correlated with the high fixation of ^18^F-DPA-714 in DIPG cells. In parametric maps showing the brain distribution of *V*_T_ values, shallow binding can be observed in the sham group, in contrast with the tumor-bearing group (Figure 4A).

## 3. Discussion

PET imaging is increasingly regarded as a reference technique for the clinical monitoring of cancer [27]. PET is being used for early diagnosis, dynamic therapeutic monitoring, and its ability to rationalize therapy selection. Over the last 10 years, PET imaging, mainly with ^18^F-FDG, provided essential information on pediatric brain tumors, impacting patient care [28]. However, ^18^F-FDG uptake failed to correlate with DIPG outcome [29]. The difficulty distinguishing the DIPG tumors from high normal brain metabolic activity is a major limitation. In the PDOX DIPG007 rat model, ^18^F-FDG PET scans did not allow for localizing the tumor. Further, there was no difference in the pons uptake of ^18^F-FDG between the tumor-bearing and the sham group. DIPG patients would benefit greatly from PET imaging with a more specific PET biomarker to improve the evaluation of treatment efficacy, especially with the numerous innovative therapies under clinical investigation. The idea is not only to diagnose but more importantly to monitor tumor response to prevent premature treatment discontinuation or on the contrary late progression diagnoses. We hypothesized that TSPO PET using ^18^F-DPA-714 may be a potent biomarker strategy, considering that TSPO PET imaging has been used in several gliomas in preclinical and clinical studies presenting high TSPO expression compared to healthy surrounding brain tissue [21,23]. TSPO is one of the hallmarks of glioblastoma and its imaging has the potential to improve patient care drastically [30]. TSPO is highly expressed in DIPG007 compared to the healthy brain tissue in vitro and in vivo after implantation (Figure 1). One difficulty in TSPO PET imaging is to distinguish tumoral TSPO expression from glial cells activation. Expression of TSPO by activated microglia and reactive astrocytes involved in neuroinflammation has been extensively described in the literature [31]. Glioma-induced neuroinflammation could impact the tumor-specificity of TSPO PET imaging [32]. Nevertheless, in our study, immunohistochemistry reveals a strong and major correlation between TSPO expression and the density of DIPG cells with low impact of microglial cells. Some activated microglial cells infiltrated the DIPG tumor (Figure 1H and Appendix A). The observation of a predominant expression of TSPO by neoplastic cells with limited contribution of GAMs to PET signal in the context of gliomas is coherent with the findings of Su et al. who lead a clinical PET and histopathologic study on 22 adults gliomas patients [33]. Lin et al. have tracked activated microglia in a representative cohort of DIPG samples by flow cytometry and immunohistochemistry. Their study compared microglia infiltration in DIPG samples with adult glioma samples [34]. They assessed that DIPG can be considered a non-inflammatory tumor compared to adult glioma in accordance with our observation.

The in vivo characterization of DIPG007 led to a clear difference in ^18^F-DPA-714 uptake in the pons between the sham group and the DIPG group 4 weeks after tumoral cell or vehicle injection, respectively. Analysis and interpretation of ^18^F-DPA-714 images can be challenging because of the basal expression of TSPO by vascular endothelial and ependymal cells [35]. Moreover, the cerebellum which is frequently used as a reference region because of its low basal expression of TSPO was infiltrated by the tumor in our PDOX DIPG model. Another reference region had to be selected to analyze our data. The striatum was chosen as a reference region since no tumor infiltration was observed. The striatum also showed the lowest ^18^F-DPA-714 uptake among all brain regions. Our work demonstrates that it is possible to quantitatively detect DIPG lesion by TSPO PET imaging. However, the delineation of the tumor remains challenging due to the heterogeneous expression of TSPO. In the pons, significant uptake of ^18^F-DPA-714 in the IV^th^ ventricle perturbates the analysis of the image. This accumulation is directly linked to TSPO expression from ependymal cells lining the ventricles [36]. The SUV, recommended for PET image analyses, presents in our case a limitation to discriminate diffuse tumor at the periphery of the tumor from background. Using the Logan plot analysis, we generated ^18^F-DPA-714 parametric *V_T_* images for the sham and the DIPG groups. Logan plots have been widely used in PET image analysis, showing an improved performance in terms of signal-to-noise discrimination, which is essential to delineate the tumor [37]. Logan *V_T_* maps revealed an expected high uptake of ^18^F-DPA-714 in the pons in the DIPG group corresponding to the tumor localization as validated by immunohistochemistry. Logan V_T_ facilitates TSPO PET image analysis, demonstrating the feasibility of this approach to assess treatment efficacy. Parametric mapping of ^18^F-DPA-714 could be envisaged as a complementary imaging modality for accurate radiotherapy dose management and may allow increasing radiation efficacy exclusively on the tumor region, sparing the healthy and highly functional brainstem region.

We concede that our study focused on a single DIPG experimental model. However, the PDOX DIPG model of our study presents the most prevalent DIPG mutations, H3F3A (K27M) and ACVR1 (R206H). Moreover, the cells were injected into the IV^th^ ventricle, which allows tumor infiltration in the brainstem, a relevant environment for DIPG Studies with different patient-derived DIPG cell lines implanted in mice or rats and studies with DIPG patient biopsies should be performed to assess the scope of our findings. Several therapeutic strategies should also be tested to confirm the use of TSPO imaging as an accurate and relevant diagnostic/therapeutic biomarker for this pathology. To our knowledge, we performed the first study on TSPO PET imaging for DIPG. Accurate biomarkers and methodology could improve DIPG patient care by assisting the therapy management. Our results may therefore pave the way for extended research on DIPG and TSPO PET imaging using dedicated radioligands.

## 4. Materials and Methods

### 4.1. Primary DIPG Cell Culture

HSJD-DIPG-007 (DIPG007) cell line was provided by Dr Angelo Montero Carcaboso (Sant Joan de Deu Hospital, Barcelona, Spain), where it was established from human DIPG tumor tissue [38]. Cells were cultured in neurospheres in DMEM-F12/Neurobasal-A (1:1, Gibco Grand Island, NY, USA) serum-deprived medium supplemented with: HEPES (10 mM, Gibco), sodium pyruvate (1 mM, Gibco), non-essential amino-acids (1%,Gibco), GlutaMax (1%,Gibco), B27-Supplement (Gibco), hEGF (20 ng·mL^−1^, Preprotech, Cranbury, NJ, USA), human bFGF (20 ng·mL^−1^, Preprotech), hPDGF-AA (10 ng·mL^−1^, Preprotech), hPDGF-BB (10 ng·mL^−1^, Preprotech), heparin (2 µg·mL^−1^, Preprotech) and antibiotic-antimycotic mix (1%, Gibco). U87 cells were cultured in DMEM supplemented with fetal bovine serum (10%, Gibco) and antibiotic-antimycotic mix (1%, Gibco). Cells were maintained in a 37 °C, 5% CO_2_ atmosphere and were periodically tested for the absence of mycoplasma contamination (MycoAlert^®^, Lonza, Basel, Switzerland).

### 4.2. Animals and Ethics

Animal experiments have been performed according to the European directive 2010/63/EU and its transposition in French law (Décret no. 2013–118). Experiments were conducted at the imaging facility CEA-SHFJ (authorization APAFIS#16293-2018072609593031/ethics committee no. 44). Athymic female nude rats RH-Foxn1^rnu^ (Envigo Laboratories, Gannat, France) were housed in standard conditions (polycarbonate cages, aspen wood as bedding material, two rats in each cage, room temperature 22 °C, humidity 40%) under a regular 12-h dark/light cycle in a ventilated cabinet. Animals were fed with standard nutritional cubes. Food and water were available ad libitum. Rats were provided nesting materials and environmental enrichments. In total, 12 rats were used in this study.

### 4.3. Orthotopic DIPG Rat Model

Intracranial tumors were established by stereotactic implantation of DIPG007 cells in the brain of 4-week-old RH-Foxn1^rnu^ rats (*n* = 8, 102 ± 8 g). Rats were anesthetized with isoflurane (3% for induction and 2% for maintenance) in O_2_. 0.05 mg.kg^−1^ of buprenorphine was subcutaneously administered at the end of the intervention to prolong analgesia. 7.5 × 10^5^ DIPG007 cells dispersed in 5 µL (70% Matrigel^®^ (Corning, Corning, NY, USA), 30% DMEM F12/Neurobasal A 1:1) were injected using a Hamilton syringe in a stereotactic apparatus (Stoelting, Wood Dale, IL, USA) into the IV^th^ ventricle. Stereotactic coordinates were as follows: 1 mm posterior and 2 mm lateral from the lambda, and 8 mm below the dura. A sham group of RH-Foxn1^rnu^ rats (*n* = 4; 81 ± 8 g) were injected with 5 µL of the cell injection medium only at the exact coordinates.

### 4.4. PET Imaging

The tumors were allowed to grow 28 days before imaging experiments. This imaging time was determined according to the tumor development kinetic described by Chaves et al. [5]. First, a 60 min dynamic brain PET scan was acquired right after the injection of N, N-diethyl-2-(2-(4-(2-^18^F-fluoroethoxy)phenyl (^18^F-DPA-714). Two days after, a 20 min static PET scan was acquired 1 h after the injection of ^18^F-2 fluoro-2-deoxy-d-glucose (^18^F-FDG).

^18^F-DPA-714 was synthesized as previously described [39]. ^18^F-FDG (Glucotep^®^) was purchased from Cyclopharma (Saint Beauzire, France). For both experimental and control sham groups, ^18^F-DPA-714 and ^18^F-FDG were intravenously injected at the dose of 169 ± 39 and 148 ± 26 kBq.g^−1^ of body weight, respectively, into the tail vein of the DIPG tumor-bearing or Sham RH-Foxn1^rnu^ rats. During all imaging sessions, rats were anesthetized with isoflurane (3.0% for induction, 1.5–2.5% for maintenance) in O_2_. Images were acquired on cross-calibrated Inveon micro PET/CT and Inveon micro PET scanners (Siemens, Knoxville, TN, USA). After each PET scan, a transmission scan or a CT scan was performed for photon attenuation correction. PET images were reconstructed with the Inveon Acquisition Workspace software (2.1) using a three-dimensional ordinary Poisson ordered-subset expectation-maximization followed by a maximum a posteriori algorithm (OP-OSEM3D-MAP). Normalization and corrections for dead-time, scatter, decay and attenuation were applied to all PET data. The 24 times frames for the dynamic scans were 3 × 30, 5 × 60, 5 × 120, 3 × 180, 3 × 240, 4 × 300, and 1 × 150 s.

### 4.5. Image Analysis

Data visualization and analysis were performed using the PMOD software (3.9). ^18^F-FDG and ^18^F-DPA-714 PET scans of the juvenile rats were all scaled up by 20% and coregistered to the W. Schiffer rat ^18^F-FDG brain PET template available in PMOD [40]. Seven volumes of interest (VOIs) were defined using the software template, i.e., the pons, medulla, cerebellum, choroid-plexus, midbrain, striatum, and the rest of the forebrain. A supplemental volume of interest was defined in the left cardiac ventricle on early time frames to generate an image-derived whole blood input function. Standardized uptake values (SUV) were calculated according to the equation:SUV = Tissue activity concentration (kBq/cm^3^)/(Injected dose (kBq)/Body weight (g)),(1)
assuming that 1 cm^3^ of the brain equals 1 g.

To compare SUVs in different VOIs, ^18^F-FDG PET scans were averaged on the 20 min of the acquisition and dynamic ^18^F-DPA-714 PET scans were averaged from 25.5 to 60 min post-injections. Volumes of distribution (*V*_T_) parametric map were established by Logan graphical analysis [41] using the dynamic ^18^F-DPA-714 PET data and an image-derived whole blood input function to intensify the contrast in ^18^F-DPA-714 distribution. Data were fitted from 22.5 min after injection for the Logan analysis based on the time course of tissue-to-blood ratios (Appendix A).

### 4.6. Immunofluorescence and Immunochemistry

After the last imaging session, rats were sacrificed, and their brains were collected, immersed in isopentane, and frozen in liquid nitrogen. Frozen tissues were stored at −80 °C. Frozen brains of tumor-bearing and sham rats were sectioned with a cryostat (Leica CM3050 S, Leica biosystems, Nussloch, Germany) in the pons and striatum areas. The sections of 10 µm were placed on adhesives slides (Superfrost Ultra Plus, Fisher Scientific, Houston, TX, USA). Autoradiography, haematoxylin/eosin (H&E), and immunofluorescent staining were performed on adjacent brain sections of the pons and striatum areas.

The autoradiographic study was performed by incubating brain sections with ^18^F-DPA-714 (41 ± 2 GBq) according to a previously described protocol [42]. Specific binding was assessed using an excess (20 µM) of unlabeled DPA-714. Slides were placed in contact with a phosphor screen in a cassette (Molecular Dynamics) for 4 h. Images were acquired at 50 µm resolution with an imager (Storm 860 Molecular Imager, Molecular Dynamics). The same slides were used for H&E staining. Slides were fixed in neutral buffer formalin 10% for 10 min, then washed with distilled water. Slides were stained with haematoxylin (Haematoxylin Harris, Sigma) and Eosin Y (Eosin Y alcoholic solution, Sigma, Saint-Louis, MO, USA) according to previously reported protocol [43].

Adjacent sections were used for immunofluorescent staining. Slides were fixed in neutral buffer formalin 10% for 15 min. Then they were washed with phosphate buffer saline (PBS). To permeabilize cell membranes, slides were immersed in methanol: acetone (1:1) and Triton 0.1% in PBS. Slides were blocked with a 5% bovine serum albumin (BSA, Sigma) solution with 0.5% Tween in PBS. Slides were incubated for 1 h with primary antibodies in the blocking solution. The first set of sections was stained for TSPO with rabbit anti-rat and human TSPO IgG (1:200, EPR5384, Novus biological, Centennial, CO, USA) and for human mitochondria with mouse anti-human mitochondria IgG (1:200, MAB1273, EMD Millipore, Burlington, MA, USA). Sections were washed three times in PBS and incubated for 30 min with secondary antibodies, Alexa Fluor 488 goat anti-rabbit IgG (1:1000, Invitrogen, Waltham, MA, USA) and Alexa Fluor 546 donkey anti-mouse IgG (1:1000, Invitrogen). The second set of slides was stained for TSPO, for CD11b with a mouse anti-rat CD11b IgG (1:200, MCA275R, AbDSerotec, Hercules, CA, USA), and for GFAP with a chicken anti rat GFAP IgG (1:500, AB4674, Abcam, Cambridge, UK). Sections were washed three times in PBS and incubated for 30 min with secondary antibodies, Alexa Fluor 488 goat anti-rabbit IgG (1:1000), Alexa Fluor 546 donkey anti-mouse IgG (1:1000), and Alexa Fluor 647 goat anti-chicken IgG (1:1000, Invitrogen). The H&E and immunofluorescent stained sections were scanned with a 20× objective using an AxiObserver Z1 microscope (Zeiss, Oberkochen, Germany). Regions of interest were scanned with a 40× objective.

### 4.7. Western Blot Analysis

Lysates from DIPG007 cells cultured as neurospheres and rat pons five weeks post-DIPG xenograft were collected in cell lysis buffer (Cell signaling, Danvers, MA, USA) supplemented with proteases/phosphatases inhibitors (Halt™ Protease and Phosphatase Inhibitor Cocktail (100×), ThermoFischer, Waltham, MA, USA). Protein concentration in each sample was measured by bicinchoninic acid assay (Bicinchoninic Acid Protein Assay Kit, Sigma-Aldrich). Denatured proteins (30 µg) were separated by gel electrophoresis (Mini Protean TGX precast gel, Biorad, Hercules, CA, USA) and transferred to the PVDF membrane. Aspecific binding sites were blocked with a 5% BSA solution. Membranes were incubated with an anti-TSPO primary antibody (1:1000, clone EPR5384, Novus biological) diluted in blocking buffer overnight at 4 °C and then incubated with a donkey anti-rabbit HRP conjugated secondary antibody (1:10,000, Jackson laboratories, Bar Harbor, ME, USA). Membrane revelation was performed with the Clarity Western Substrate (Biorad). Chemiluminescence was measured with a Fusion Fx (Vilber).

### 4.8. Statistics

All statistical analyses were performed in R (v.4.0.2) [44], and the confidence level associated was fixed at 95%. The influence of the DIPG on radiotracers’ uptake was assessed with an ANOVA with two qualitative variables: the brain structure and the rat group. This variance analysis was followed by pairwise comparison of means with Bonferroni’s *p*-value adjustment within brain structures between rat groups.

## Figures and Tables

**Figure 1 ijms-23-12476-f001:**
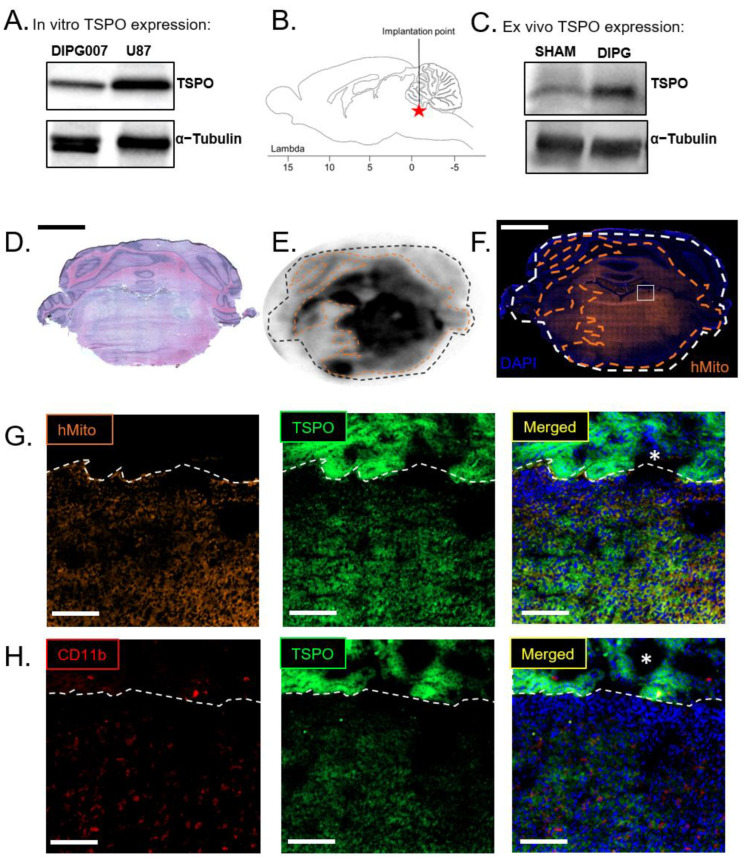
In vitro and ex vivo TSPO expression of HSJD-DIPG-007 cells. (**A**) In vitro DIPG007 and U87 TSPO expression as revealed by Western Blot. (**B**) Implantation point to establish the orthotopic DIPG murine model. (**C**) Ex vivo TSPO expression in the back brain of sham and DIPG007 cell-bearing rats as revealed by western blot. Western blot quantifications are available in Appendix A. (**D**,**E**) Hematoxylin/eosin staining and ^18^F-DPA-714 autoradiography of the same brain section. (**F**) Human mitochondria immunostaining on an adjacent section. The orange contour is the area where fixation of ^18^F-DPA-714 on the autoradiography is observed. (**G**) TSPO and human mitochondria immunostaining in a tumoral area near the IVth ventricle indicated by the white box on F. (**H**) TSPO and CD11b immunostaining in a tumoral area near the IVth ventricle. Stars (*) indicates the IV^th^ ventricle ependymal cells. Scale bars = 2.5 mm (**D**,**F**) and 150 µm (**G**,**H**).

**Figure 2 ijms-23-12476-f002:**
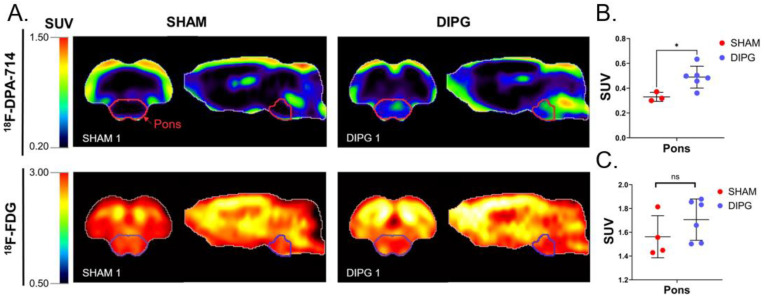
^18^F-FDG and ^18^F-DPA-714 uptakes in the brain of DIPG-bearing and sham rats. (**A**) SUV-normalized brain PET images of ^18^F-DPA-714 and ^18^F-FDG uptakes in one sham (SHAM 1) and one tumor-bearing rat (DIPG 1). (**B**) Mean ^18^F-DPA-714 SUV in the pons. (**C**) Mean ^18^F-FDG SUV in the pons (*n* = 4 in the sham group and *n* = 6 in the DIPG group). Statistical significance was determined using *t*-test with * *p* < 0.05. Data are represented as mean ± SD.

**Figure 3 ijms-23-12476-f003:**
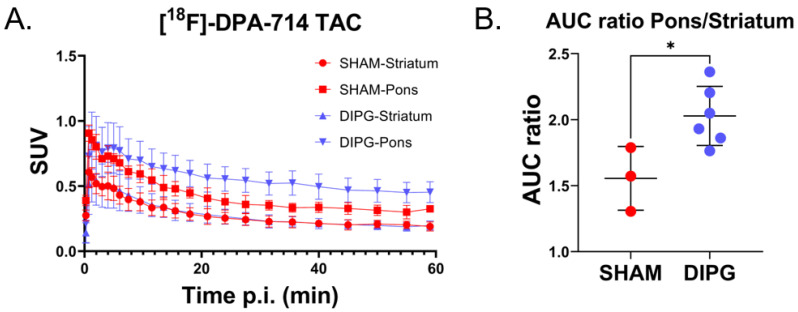
^18^F-DPA-714 uptakes in the pons and the striatum over time post injection. (**A**) Time activity curves (TAC) of ^18^F-DPA-714 in different brain structures of sham and tumor-bearing rats. (**B**) Individual area under the curves (AUC) pons to striatum ratio. Statistical significance was determined using *t*-test with * *p* < 0.05. Data are represented as individual data and mean ± SD.

**Figure 4 ijms-23-12476-f004:**
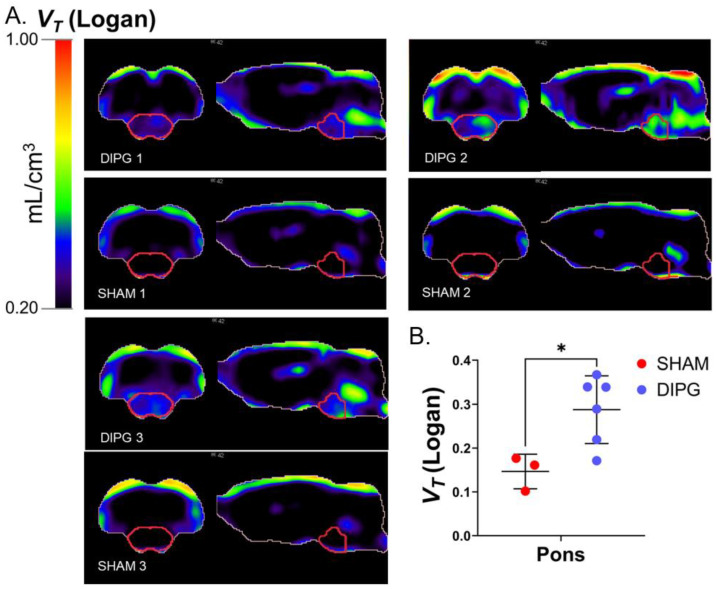
Distribution volume of ^18^F-DPA-714. (**A**) Distribution parametric brain images of three sham and tumor-bearing rats. (**B**) Mean distribution volumes of in the pons^18^F-DPA-714. Total volumes of distribution were estimated using the Logan graphical analysis. Statistical significance was determined using *t*-test with * *p* < 0.05. Data are represented as mean ± SD.

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
