# Peer review of "TSPO PET Imaging as a Potent Non-Invasive Biomarker for Diffuse Intrinsic Pontine Glioma in a Patient-Derived Orthotopic Rat Model"

_ijms, 2022, doi:10.3390/ijms232012476_

Round 1
Reviewer 1 Report
Thanks a lot for your invitation to review this manuscript "TSPO PET imaging as a potent non-invasive biomarker for Diffuse Intrinsic Pontine Glioma in a patient-derived orthotopic rat model" (ijms-1961988). The obtained data confirm each other and the experimental design of this work is well arranged. Their findings are exciting. I read their original version of the manuscript very carefully and think this work could be acceptable in its current form.
Author Response
It is a real pleasure to read the positive comments on our manuscript. We are very happy for our work meets the expectation and arouse the interest of the reviewer.

Reviewer 2 Report
In this study, authors validated the use of TSPO PET imaging as a potential biomarker evaluating the extent of DIPG tumors in rats. The study was largely well done and convincing. There are some minor typos in the supplementary figure legends that should be corrected. For western blots and immunofluorescent staining results, please also present the quantification of the results. Please add DAPI staining to the merged pictures of all IF staining. For all figure legends, please add information of statistical analysis and exact replicates of each experiment. As author stated in the limitations, this experiment was limited to one model in rats. Therefore, please also discuss the potential use of this approach in other models.
Author Response
We would like to thanks the reviewer for the different comments and remarks.
To answer the different questions, please find enclosed point by point our responses:
“There are some minor typos in the supplementary figure legends that should be corrected.”.
- We have corrected all the typos we have encountered in the supplementary data. All the corrections were performed in track changes mode.
“For western blots and immunofluorescent staining results, please also present the quantification of the results.”.
- We have quantified the western blots dots using imageJ. We have added this quantification on the supplementary file. For the immunofluorescent staining, we are using an epifluorescence microscope that render difficult a correct quantification of the labeling. We have not used confocal microscope that could be quantified unfortunately. Our point using epifluorescence images was more focused to assess the cells that expression TSPO, and to demonstrate that TSPO can be related more to the tumor cells than the glial cells.
“Please add DAPI staining to the merged pictures of all IF staining.”.
- We have completed all the merged pictures to present separately the DAPI staining as suggested by the reviewer.
“For all figure legends, please add information of statistical analysis and exact replicates of each experiment.”.
- We have corrected the figure legends by adding the statistical analysis and the replicated of each experiment.
“As author stated in the limitations, this experiment was limited to one model in rats. Therefore, please also discuss the potential use of this approach in other models.”.
- The question of presenting only one model is relevant. To obtain one patient-derived xenograft model is challenging since the cells implanted must represent with accuracy the human model. That’s why we expect that our study will motivate other research groups to perform and strengthen our purpose. We expect that our study will be the first stone to gather several data with TSPO-PET imaging on such terrible pathology. We believe strongly that our approach can help patient care and potential tumor remission. We have consolidated our discussion by insisting as suggested by the reviewer about this point by modifying the discussion with: “Studies with different patient-derived DIPG cell lines implanted in mice or rats and studies with DIPG patient biopsies should be performed to assess the scope of our findings. Several therapeutic strategies should also be tested to confirm the use of TSPO imaging as an accurate and relevant diagnostic/therapeutic biomarker for this pathology.”
